# A Diffusion-Driven Fine-Grained Nodule Synthesis Framework for Enhanced Lung Nodule Detection from Chest Radiographs

**Shreshtha Singh**[*1]                                                      SHRESHTHA.SINGH@QURE.AI
**Aryan Goyal**[*1,2]                                                           21D180006@IITB.AC.IN
**Ashish Mittal**[*1]                                                         ASHISH.MITTAL@QURE.AI
**Manoj Tadepalli**[1]                                                     MANOJ.TADEPALLI@QURE.AI
**Piyush Kumar**[1]                                                         PIYUSH.KUMAR@QURE.AI
**Preetham Putha**[1]                                                     PREETHAM.PUTHA@QURE.AI

[1] *Qure.ai, India*

[2] *Indian Institute of Technology, Bombay*

**Editors:** Accepted for publication at MIDL 2026

## Abstract

Early detection of lung cancer in chest radiographs (CXRs) is crucial for improving patient outcomes, yet nodule detection remains challenging due to their subtle appearance and variability in radiological characteristics like size, texture, and boundary. For robust analysis, this diversity must be well represented in training datasets for deep learning based Computer-Assisted Diagnosis (CAD) systems. However, assembling such datasets is costly and often impractical, motivating the need for realistic synthetic data generation. Existing methods lack fine-grained control over synthetic nodule generation, limiting their utility in addressing data scarcity. This paper proposes a novel diffusion-based framework with low-rank adaptation (LoRA) adapters for characteristic controlled nodule synthesis on CXRs. We begin by addressing size and shape control through nodule mask conditioned training of the base diffusion model. To achieve individual characteristic control, we train separate LoRA modules, each dedicated to a specific radiological feature. However, since nodules rarely exhibit isolated characteristics, effective multi-characteristic control requires a balanced integration of features. We address this by leveraging the dynamic composability of LoRAs and revisiting existing merging strategies. Building on this, we identify two key issues: overlapping attention regions and non-orthogonal parameter spaces. To overcome these limitations, we introduce a novel orthogonality loss term during LoRA composition training. Extensive experiments on both in-house and public datasets demonstrate improved downstream nodule detection. Radiologist evaluations confirm the fine-grained controllability of our generated nodules, and across multiple quantitative metrics, our method surpasses existing nodule generation approaches for CXRs.

**Keywords**: Lung Nodule Synthesis, Chest Radiograph, Diffusion Models , LoRA , LoRA Merging

## 1. Introduction

Lung cancer remains a leading cause of cancer mortality (Bray et al., 2018). Early detection is critical and localized tumors can reach five-year survival rates above 70% (see, 2017).

---

[*] Contributed equally

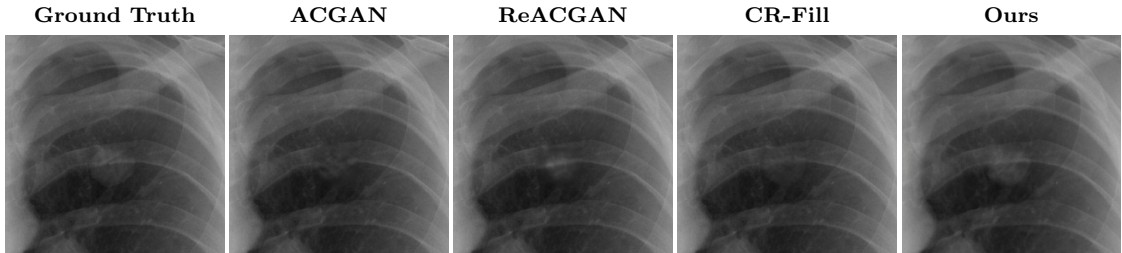

Figure 1: Comparison of nodule generations across different methods: ACGAN(Odena et al., 2017), ReACGAN(Lee et al., 2021), CR-Fill(Zhao et al., 2021), and Ours.

Despite advances in imaging, many cases are first suspected on chest X-rays (CXR), one of the most common tests (Rogers et al., 2010); accurate nodule detection on CXR is therefore essential. Pulmonary nodules exhibit key characteristics such as size, calcification, border definition, and homogeneity that are essential for malignancy assessment (Albert and Russell, 2009). Robust CAD requires datasets spanning these traits, yet high-quality annotated CXRs are scarce, manual labeling is labor-intensive, costly, and suffer from intra-observer variability (Irvin et al., 2019). Synthetic augmentation can expand training data and improve detectors (Schultheiss et al., 2021; Hanaoka et al., 2024; Wang et al., 2022), but current nodule synthesis or inpainting approaches lack fine-grained control over nodule characteristics. Given the high CXR miss rate, precise and clinically faithful control of nodule attributes is needed.

Diffusion models offer strong potential for precise generative control, outperforming GANs in image quality, stability, and diversity (Dhariwal and Nichol, 2021; Yang et al., 2022). Among them, Diffusion Transformers (DiT) (Peebles and Xie, 2023) achieve superior performance compared to large UNet-based architectures such as SDXL (Podell et al., 2023). Controllability in diffusion models has advanced through lightweight conditioning mechanisms, including ControlNet (Zhang et al., 2023) and LoRA-based methods (Hu et al., 2021; Koh et al., 2023; Lyu et al., 2023), which steer generation toward specific attributes without full-model retraining. Concept Sliders (Gandikota et al., 2023) further enable continuous and compositional control by learning semantic directions in latent activation space.

In our work, we adopt DiT as our backbone and condition it with binary masks to control nodule shape, size, and location. After training the backbone, we attach separate LoRA modules (Hu et al., 2021) for four radiological attributes: calcification, border definition (regular/irregular), homogeneity, and subtlety by using characteristic-specific subsets to capture fine-grained distinctions without affecting general nodule synthesis. Since nodules often exhibit multiple attributes, we explore LoRA combination strategies like LoRA-Switch (Kong et al., 2024), linear merging (Prabhakar et al., 2024) and training-based fusion Zi-pLoRA (Shah et al., 2023). We find these methods limited by spatial competition and interference from non-orthogonal adapter weights. To address this, we propose a training-based merging strategy with a Frobenius norm penalty that encourages orthogonality across LoRA matrices.

The proposed framework (i) introduces a novel diffusion-based approach for generating synthetic lung nodules on CXRs, (ii) enables controllable synthesis of key radiological

characteristics through characteristic-specific and merged LoRA adapters, and (iii) is validated through extensive experiments on both in-house and public datasets, demonstrating consistent improvements over existing methods, including downstream detection gains with AUCs of 0.90 on JSRT and 0.93 on CheXray14, further supported by radiologist evaluations. The code will be released at: https://github.com/shreshthasingh00/Nodule-Crafter-Diffusion-driven-Nodule-synthesis-on-CXR/

## 2. Related Work

**Lung Nodule Synthesis:** Early work synthesized nodules by forward-projecting CT-derived annotations onto radiographs (Schultheiss et al., 2021; Litjens et al., 2010; Behrendt et al., 2023). Other methods generate nodules directly in masked CXR regions using inpainting (Sogancioglu et al., 2018) or feature-level blending (Gündel et al., 2021). GAN-based approaches (Shen et al., 2022) enable factorized control over shape, size, and texture but lack fine-grained characteristic manipulation. To the best of our knowledge, no prior work has investigated diffusion models for fine-grained, controllable nodule synthesis in CXRs.

**Controllable Image Generation:** Early controllability in diffusion models was achieved through classifier and classifier-free guidance (Dhariwal and Nichol, 2021; Ho and Salimans, 2022), followed by lightweight conditioning modules such as T2I-Adapters and ControlNet (Mou et al., 2023; Zhang et al., 2023). Editing and personalization methods, including prompt-to-prompt and DreamBooth (Hertz et al., 2022; Ruiz et al., 2023), enabled localized and concept-specific control. Among these, LoRA (Hu et al., 2021) has emerged as a dominant mechanism for controllable generation as it enables efficient low-rank fine-tuning and allows for composability of adapter modules

**LoRA Merging:** Merging multiple LoRA adapters remains challenging, with works such as ZipLoRA, Mix-of-Show, and K-LoRA (Shah et al., 2023; Gu et al., 2023; Ouyang et al., 2025) showing that naive fusion leads to concept conflicts, loss of identity, and attenuation of fine details. Several approaches aim to mitigate these issues: DO-Merging (Zheng et al., 2025) enforces layer-wise orthogonalization of LoRA directions, LoRI (Zhang et al., 2025) reduces cross-task interference via sparse masking and frozen projections, and ZipLoRA further introduces trainable merger coefficients to balance layer-wise adapter contributions. These methods attempt to resolve conflicts after independent trainings. In contrast, our method integrates a Frobenius norm based orthogonality loss directly into the training of each characteristic-specific adapter, ensuring that the learned LoRAs are inherently compatible for merging.

## 3. Datasets

**Nodule Characteristics Definitions:** Pulmonary nodules exhibit several radiological attributes important for distinguishing them from mimickers and assessing malignancy. Their size ranges from a few millimeters up to 3 cm. Calcification, arising from calcium deposits, is frequently associated with benignity (Khan et al., 2010). Border definition reflects edge morphology: regular, well-defined margins are typically stable, whereas irregular, spiculated, or lobulated borders may indicate malignancy (Zhang et al., 2020). Homogeneity describes texture uniformity; homogeneous nodules show consistent intensity, while inho-

| (a) Original patch | (b) Nodule mask | (c) Generated Patch |
|---|---|---|

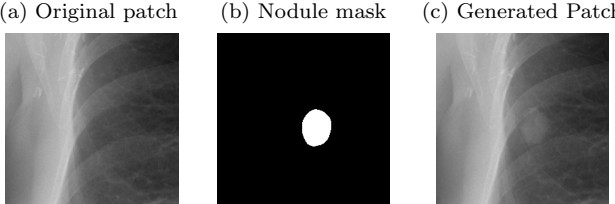

Figure 2: Diffusion Backbone generation on a chest X-ray patch: given an original CXR patch and a binary nodule mask, the model generates a nodule within the masked region

mogeneous ones exhibit variation due to necrosis or vascularity, features often linked to malignant processes (Balagurunathan et al., 2019). Perceptual subtlety is also critical, as nodules may be faint or obscured by ribs and vessels, making detection challenging.

**In-house and Public Datasets:** Our in-house dataset comprises 1.2M frontal-view CXRs from partner hospitals, including 40k chest X-rays with pulmonary nodules. Each nodule is delineated with shape annotations and labeled for calcification (7,875), regular border (10,424), irregular border (5,153), homogeneous texture (4,640), inhomogeneous texture (5,883), and subtlety (5,000 cases graded 1–5), with all annotations independently provided by three experienced radiologists. We further split the in-house trainset for generation and downstream trainings. For evaluation, we additionally use the public *ChestX-ray14* (Kufel et al., 2023) and *JSRT* (Shiraishi et al., 1996) datasets, where JSRT also includes subtlety scores (1 = most subtle, 5 = most obvious) for nodules. Both public datasets provide nodule bounding boxes; we refine these using our segmentation model trained on the in-house data and align predictions with the provided boxes. All selected segmentations were reviewed by radiologists for consistency. Dataset statistics for all the datasets are provided in Table 1.

Table 1: Summary of datasets used for training and evaluation.

|  | In-house | | | JSRT | CheX-ray14 |
|---|---|---|---|---|---|
|  | *Diffusion Trainset* | *Downstream Trainset* | *Testset* | *Testset* | *Testset* |
| **Total samples** | 1,100,000 | 100,000 | 12,000 | 247 | 500 |
| **Nodule samples** | 28,000 | 10,000 | 2,000 | 154 | 66 |

## 4. Methodology

### 4.1. Background

**Diffusion Models** are a class of generative models that learn to approximate complex data distributions by iteratively transforming random noise into structured samples. They operate through a two-phase process: a *forward diffusion process*, in which Gaussian noise is incrementally added to training data over a sequence of steps, and a *reverse denoising process*, where a network is trained to recover the original data by gradually removing the noise. This learned denoising procedure allows the model to sample from the target distribution by reversing the noising trajectory. The forward diffusion process is modeled as a Markov chain, where Gaussian noise is added at each time step. Given an initial data

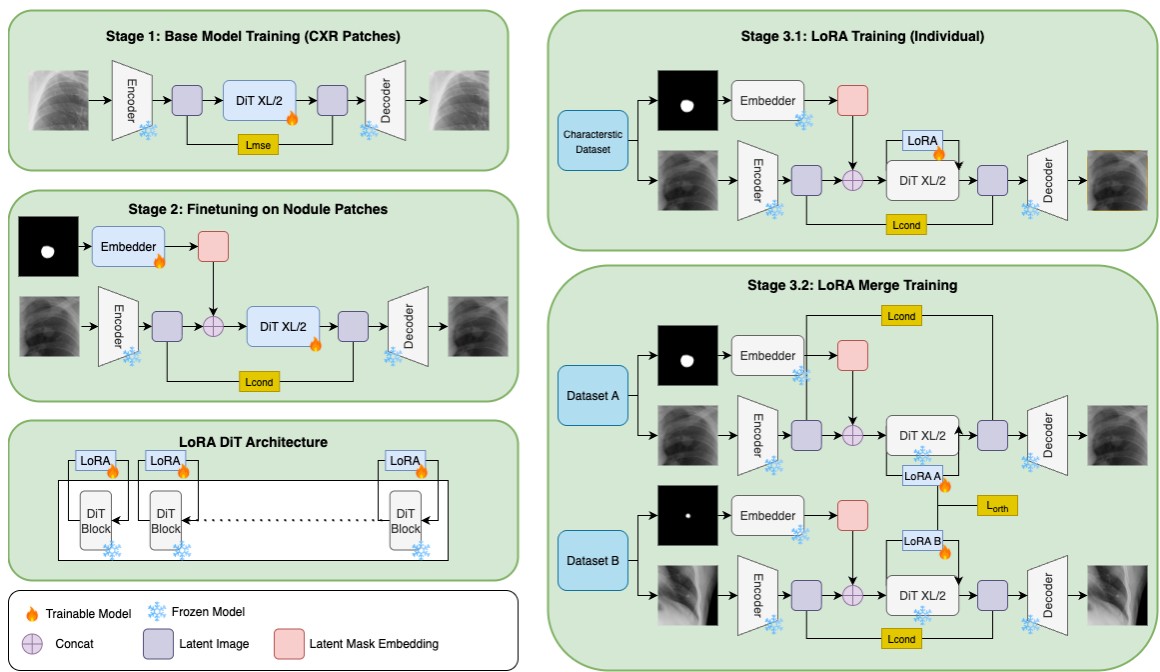

Figure 3: Overview of the pipeline with characteristic control. Training proceeds in 3 stages: **Stage 1 (Base Model Training)**-pre-train DiT-XL/2 on 2M of CXR patches; **Stage 2 (Nodule Patch Fine-tuning)**-finetune on nodule-centered patches with binary nodule masks for localized synthesis; **Stage 3.1 (Characteristic-Specific LoRA Training)**-train individual LoRA adapters (calcification, homogeneity, border regularity, subtlety) with the backbone frozen, using characteristic-curated datasets; **Stage 3.2 (LoRA Merge Training)**-jointly train selected adapters with an orthogonality regularizer

sample $x_0$, the noised sample at time step $t$ is computed as:

$$q(x_t|x_{t-1}) = \mathcal{N}(x_t; \sqrt{1 - \beta_t}x_{t-1}, \beta_t I) \tag{1}$$

where $\beta_t$ is the noise schedule controlling the amount of noise added at each step. The final state $x_T$ approaches a pure Gaussian noise distribution. To train the model, we optimize:

$$L_{\text{cond}} = \mathbb{E}_{x_0,t,\epsilon,c} \left[\|\epsilon - \epsilon_\theta(x_t, t, c)\|^2\right] \tag{2}$$

where $\epsilon_\theta(x_t, t, c)$ represents the model's prediction of the noise at time $t$ and condition $c$.

## 4.2. Diffusion Backbone

We build our framework on a Diffusion Transformer leveraging its capacity for large scale data and high-fidelity generation. We first pre-train DiT-XL/2 on approximately 2 million unlabeled chest radiograph patches, enabling the model to capture the semantic structure of thoracic anatomy and generate realistic CXR patches. To specialize the model for localized nodule synthesis, we finetune it on curated nodule patches using binary masks as spatial

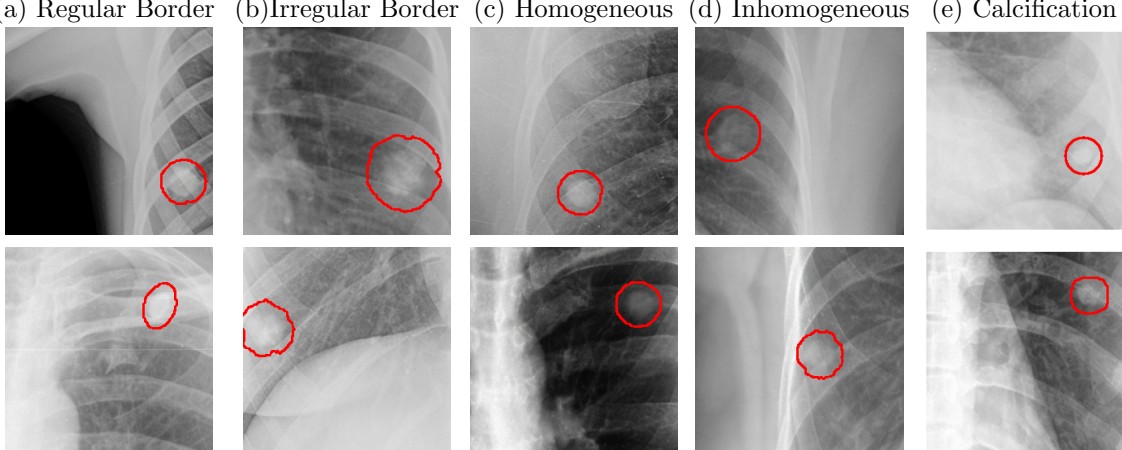

(a) Regular Border  (b)Irregular Border  (c) Homogeneous  (d) Inhomogeneous  (e) Calcification

Figure 4: Results of characteristic-specific LoRA training: (a) nodule with a regular border, (b) nodule with an irregular border, (c) nodule with homogeneous texture, (d) nodule with inhomogeneous texture, and (e) nodule with calcification.

conditioning signals. These masks define the target region for nodule placement, allowing the model at inference to synthesize nodules at desired locations through inpainting while preserving surrounding lung structures as shown in Figure 2. To further improve adherence to the conditioning masks, we apply classifier-free guidance (CFG) during inference.

**Rationale for Separate LoRA Adapters:** Using CFG to control multiple semantic labels simultaneously competes with mask-based conditioning and degrades boundary fidelity. Empirically, applying CFG to both masks and labels produced suboptimal results. Consequently, we adopt LoRA modules for characteristic control as they are computationally inexpensive while using CFG for mask conditioning. A comparison between CFG-based label control and our separate LoRA approach is provided in Appendix E.3.

### 4.3. Characteristic-Specific LoRA Adapters

We design characteristic-specific LoRA adapters for clinically relevant nodule attributes. LoRA freezes the pre-trained weights and learns a compact set of rank-decomposed updates, significantly reducing the number of trainable parameters. Given a pre-trained weight matrix $W_0$, LoRA parameterizes its update as $\Delta W = AB$, where $A \in \mathbb{R}^{d \times r}$ and $B \in \mathbb{R}^{r \times k}$, with $r \ll \min(d, k)$. Each characteristic LoRA adapter is trained on samples curated for its respective characteristic, while the DiT-XL/2 backbone remains frozen. Qualitative results for individual characteristics are presented in Figure 4. For the subtlety attribute, we extend the Concept Sliders (Gandikota et al., 2023) approach to leverage the graded annotations in our dataset. Radiologists labeled each nodule on a discrete scale from 1 (most subtle) to 5 (most obvious). To incorporate this into training, we map the annotation level directly to the LoRA scale parameter ($\alpha$). Nodules with lower subtlety scores are assigned smaller $\alpha$ values, yielding weaker updates and less visible nodules, whereas higher scores correspond to larger $\alpha$ values, amplifying the updates and making nodules more apparent.

$$W = W_0 + \alpha(s) \, \Delta W \tag{3}$$

(a) Subtlety ($\alpha$=24)  (b) Subtlety ($\alpha$=16)  (c) Subtlety ($\alpha$=12)

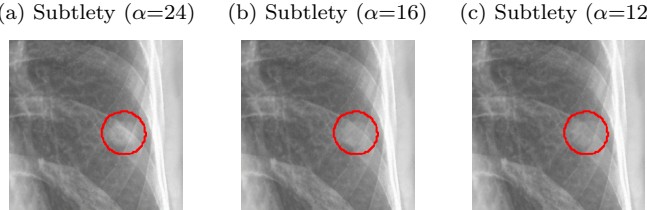

Figure 5: Subtlety slider generations on a CXR patch. From left to right: decreasing LoRA scale $\alpha$ values produce more subtle nodules.

where $s \in \{1, 2, 3, 4, 5\}$ is the radiologist-assigned subtlety score, and we take $\alpha(s) = 2^{2+s}$ as the LoRA scale which we found to work well empirically during subtlety slider training. During inference, we set $\alpha$ for generating nodules at different subtlety levels as shown in Figure 5.

### 4.4. Challenges in Multi-Characteristic Synthesis

Conventional merging strategies such as linear merge and switching (Zhong et al., 2024) assume that LoRA updates combine uniformly across layers. In practice, this leads to suboptimal synthesis like one characteristic often dominates and artifacts often appear near mask boundaries. In general, LoRA weights are highly sparse (60–70% near zero; $|w| < 0.01$), meaning a small subset of parameters drives perceptual change, making naive averaging brittle (Ouyang et al., 2025). Two factors contribute most: **overlapping attention regions** and **non-orthogonal updates**. Unlike natural-image settings where attributes may be spatially distinct, all adapters operate on the same nodule region, causing competing updates. Our Frobenius-norm orthogonality analysis (Nakayama, 1952) further shows that independently trained adapters are not well separated, leading to correlated updates. This non-orthogonality causes interference regardless of the merging strategy employed.

### 4.5. Orthogonality-Constrained Adapter Merging

To reduce interference, we promote orthogonality between adapter subspaces during training. For adapters with updates $W_1$ and $W_2$, we add

$$\mathcal{L}_{\text{orth}} = \|W_1 W_2^\top\|_F^2 \tag{4}$$

and apply the term pairwise when using more than two adapters. Although the formulation naturally extends to merging multiple characteristics, in this work, we evaluate the setting of two-attribute combinations. This loss penalizes correlations between the parameter subspaces of different adapters. We scale this term in the final loss using coefficient $\lambda$, allowing a trade-off between orthogonality and reconstruction fidelity. With this constraint, adapters compose reliably: linear averaging suffices, and each characteristic remains controllable via its scalar $\alpha$. As illustrated in Figure 7, our merging follows the nodule mask better preserving its structure and follows the characteristics better than just inference time linear merging. We also analyse the orthogonality across 28 transformer layers shown in Figure 6, which shows that with cross training the Frobenius norms are much lower compared to separately trained adapters.

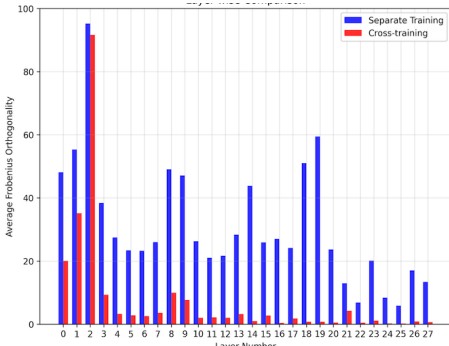

Figure 6: Layer-wise Frobenius norm comparison of cross-trained vs. separately trained adapters across two characteristic pairs.

## 5. Results and Analysis

We evaluate our framework along three criteria: (i) radiologist evaluation, (ii) quantitative assessment on standard image metrics, (iii) downstream impact on nodule detection models

### 5.1. Radiologist Evaluation

**Task 1 (Realism)** We presented a balanced set of 50 nodules (25 real and 25 generated by our base diffusion model) to three expert radiologists. On average, **90%** of real nodules were correctly identified as real, while **80%** of synthetic nodules were also labeled as real, indicating that our model produces nodules that are highly realistic and often indistinguishable from genuine cases.

**Task 2 (Controllability)** To evaluate characteristic-specific LoRA adapters, we generated 10 nodules per target feature (e.g., border type, texture) and asked 3 radiologists to verify the intended trait. Table 2 summarizes the majority-agreement rates across features.

**Task 3 (Subtlety)** We evaluated Subtlety LoRA by generating 20 synthetic nodule patches from the same mask, each rendered at 3 different levels, as shown in Figure 5. Radiologists were asked to arrange the samples in order, from the most obvious to the most subtle nodules. Across cases, majority consensus ordering aligned with our subtlety scale in 80% of cases, confirming a clear, clinically relevant progression in subtlety of generated nodules.

Table 2: Radiologist evaluation of characteristic-specific LoRA modules.

| Nodule characteristic | Agreement (%) |
|---|---|
| Calcification | 80 |
| Regular border | 90 |
| Irregular border | 100 |
| Homogeneous texture | 90 |
| Inhomogeneous texture | 100 |

### 5.2. Downstream Evaluation

**Diffusion Baseline Evaluation:**  We evaluated the detection performance of models augmented with synthesized nodules from our baseline diffusion model. A Swin-Tiny (Liu

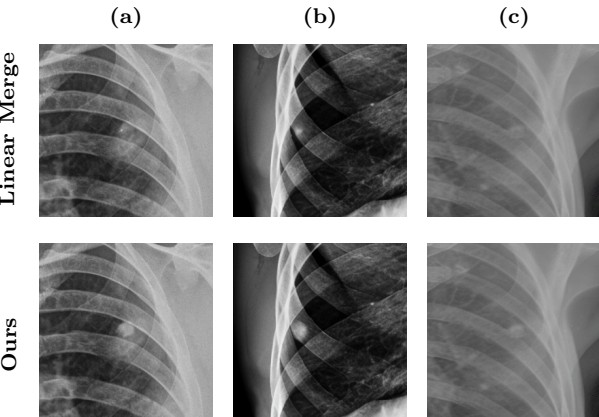

Figure 7: Qualitative comparison between Linear Merge and our orthogonal merging across three nodule-attribute configurations: (a) calcified + irregular, (b) calcified + homogeneous, (c) regular + homogeneous.

et al., 2021) encoder combined with a U-Net++ (Zhou et al., 2018) architecture was trained jointly for nodule classification and segmentation. The training set consisted of approximately 10k real nodules with nearly 100k normal CXRs, supplemented with synthesized nodules.

As shown in Table 3, augmenting the training data with our synthetic nodules consistently improved both classification and segmentation performance across all test sets. These results highlight two important trends: (1) augmenting with synthetic nodules consistently boosts downstream detection performance across datasets, and (2) performance improvements plateau or slightly decline beyond an optimal level of augmentation, suggesting that carefully balanced integration of synthetic data maximizes its effectiveness.

Table 3: Quantitative evaluation of the effectiveness of different quantities of synthesized nodule data. Reported metrics include AUC and best IoU on three test sets.

| Train Data | In-house | | JSRT | | CheX-ray14 | |
|---|---|---|---|---|---|---|
| | AUC | IoU | AUC | IoU | AUC | IoU |
| 10k real | 0.9705 | 0.3090 | 0.8560 | 0.2475 | 0.9008 | 0.5285 |
| 10k real + 2k syn | 0.9788 | 0.3222 | 0.8639 | 0.2743 | 0.9168 | 0.5293 |
| 10k real + 4k syn | 0.9780 | 0.3197 | 0.8780 | 0.2589 | 0.9245 | 0.5500 |
| 10k real + 6k syn | **0.9802** | **0.3247** | 0.8940 | 0.2894 | 0.9315 | 0.5750 |
| 10k real + 8k syn | 0.9796 | 0.3274 | 0.8864 | 0.2923 | **0.9341** | **0.5954** |
| 10k real + 10k syn | 0.9801 | 0.3056 | **0.9023** | **0.3091** | 0.9318 | 0.5613 |

**Characteristic-Specific LoRA Adapters Evaluation:** We trained a Swin-Tiny (Liu et al., 2021) encoder with a multi-head classification module for all radiological characteristics, augmenting the training set with approximately 400 synthetic nodules per characteristic. The results in Table 4 show that with augmentation the IoU score has improved across all characteristics on our in-house testset. We also evaluate the subtlety on JSRT subtlety dataset, results are provided in Appendix E.1

Table 4: Comparison of models against IoU scores trained with 5k real nodules versus 5k real nodules with 2k characteristic specific synthetic nodules across radiological features.

| Characteristic | 5k Real | 5k Real + 2k synthetic |
|---|---|---|
| Nodule | 0.2696 | **0.3002** |
| Calcification | 0.2879 | **0.3199** |
| Regular Border | 0.2941 | **0.3301** |
| Irregular Border | 0.2733 | **0.3080** |
| Homogeneous | 0.2695 | **0.3050** |
| Inhomogeneous | 0.2706 | **0.2963** |

## 5.3. Comparison with Existing Methods

To ensure a fair comparison, all baselines were trained on the same in-house dataset. We benchmark three families of generative approaches: GAN-based models, fill-based inpainting, and our Stage-2 diffusion framework (Figure 3). For inpainting, we include CR-Fill (Zhao et al., 2021), the top performer in the NODE21 Generation Track (Sogancioglu et al., 2024), given its strong CXR inpainting performance. For GANs, we evaluate AC-GAN (Odena et al., 2017) and ReACGAN (Lee et al., 2021), widely used class-conditional frameworks. To assess the impact of synthetic nodules, we augmented the training data with 10k generated samples from each method and measured classification AUC on JSRT and ChestX-ray14 (Table 5). Although all augmentations improved over using 10k real samples alone, diffusion-based augmentation achieved the highest gains of 0.9023 AUC on JSRT and 0.9318 on ChestX-ray14, demonstrating its effectiveness for downstream detection.

Table 5: Comparison of effect of synthetic-data augmentation on nodule AUC scores across *ChestX-ray14* and *JSRT*.

| Augmentation | JSRT | ChestX-ray14 |
|---|---|---|
| 10k real | 0.8560 | 0.9008 |
| 10k real + 10k ACGAN | 0.8780 | 0.9281 |
| 10k real + 10k ReACGAN | 0.8808 | 0.9259 |
| 10k real + 10k CR-Fill | 0.8786 | 0.9296 |
| 10k real + 10k DiT-XL/2(Ours) | **0.9023** | **0.9318** |

## 6. Conclusion

We introduced a novel diffusion-based framework for pulmonary nodule synthesis with characteristic-specific LoRA adapters, and an orthogonality constrained LoRA merging strategy. Experiments show that our method generates realistic and controllable nodules, outperforms GAN and inpainting-based baselines, and improves downstream CAD performance, with radiologist evaluations confirming clinical plausibility. Limitations include difficulty with some out-of-distribution generations by composition of LoRAs. Future work includes extending the merging strategy to more than two characteristics.

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

# Appendix A. Pseudocode

## A.1. Training Pipeline for DiT backbone

---
**Algorithm 1** Training Pipeline for Mask-Conditioned DiT Backbone

---
1: **Input:** Dataset $\mathcal{D} = \{(x, m)\}$ where $x$ is a CXR patch and $m$ is a binary nodule mask.
2: **Initialize:** VAE encoder–decoder, DiT backbone $f_\theta$, diffusion schedule $\{\alpha_t\}_{t=1}^{T}$, optimizer.
3: **Repeat until convergence:**
4:     Sample $(x, m) \sim \mathcal{D}$.
5:     $z_0 \leftarrow$ VAE.encode($x$).
6:     $c \leftarrow$ process($m$).
7:     Sample $t \sim \{1, \ldots, T\}$ and $\epsilon \sim \mathcal{N}(0, I)$.
8:     Form noisy latent $z_t \leftarrow \sqrt{\alpha_t}\, z_0 + \sqrt{1 - \alpha_t}\, \epsilon$.
9:     Predict noise $\hat{\epsilon} \leftarrow f_\theta(z_t, t, c)$.
10:     Compute loss $\mathcal{L}_{\text{diff}} \leftarrow \|\epsilon - \hat{\epsilon}\|^2$.
11:     Update parameters $\theta \leftarrow \theta - \eta \nabla_\theta \mathcal{L}_{\text{diff}}$.

---

## A.2. Training Pipeline for LoRA Adapter

---
**Algorithm 2** LoRA Adapter Training

---
1: **Freeze:** backbone parameters $\theta$; train LoRA parameters $\phi$ only
2: **Repeat until convergence:**
3:     Sample $(x, m) \sim \mathcal{D}$
4:     $z_0 \leftarrow$ VAE.encode($x$)
5:     $c \leftarrow$ process($m$)
6:     Sample $t \sim \{1, \ldots, T\}$ and $\epsilon \sim \mathcal{N}(0, I)$
7:     $z_t \leftarrow \sqrt{\alpha_t}\, z_0 + \sqrt{1 - \alpha_t}\, \epsilon$
8:     $\hat{\epsilon} \leftarrow f_{\theta,\phi}(z_t, t, c)$
9:     Compute $\mathcal{L}_{\text{diff}}$ (plus optional $\mathcal{L}_{\text{ortho}}$, $\mathcal{L}_{\text{con}}$)
10:     Update LoRA parameters $\phi$ using the combined loss

---

## A.3. Inference Algorithm (Conditional Sampling)

---

**Algorithm 3** Inference via Mask-Conditioned Reverse Diffusion

---

1: **Input:** Binary mask $m$, trained backbone $f_\theta$, VAE decoder, diffusion schedule $\{\alpha_t, \sigma_t\}_{t=1}^{T}$
2: Sample $z_T \sim \mathcal{N}(0, I)$
3: $c \leftarrow \text{process}(m)$
4: **For timesteps $t = T$ down to 1:**
5:     $\hat{\epsilon} \leftarrow f_\theta(z_t, t, c)$
6:     $\mu_\theta \leftarrow \frac{1}{\sqrt{\alpha_t}} \left( z_t - (1 - \alpha_t)\hat{\epsilon} \right)$
7:     Sample $z_{t-1} \sim \mathcal{N}(\mu_\theta, \sigma_t^2 I)$
8: $\hat{x} \leftarrow \text{VAE.decode}(z_0)$
9: **Output:** Synthesized CXR patch $\hat{x}$

---

# Appendix B. Experimental Setup

**Base Diffusion Model & Training Setup:** The DiT-XL/2 backbone contains 28 transformer blocks, each composed of attention and MLP components. We initialize the DiT-XL/2 backbone from a publicly available pre-trained checkpoint. Full-resolution chest X-rays are standardized to $960 \times 960$ pixels, from which $256 \times 256$ nodule-centered patches are extracted. These patches are encoded into $32 \times 32$ latent representations using the StabilityAI VAE-FT-EMA. The diffusion process follows a 1000-step DDPM with a linear noise schedule, selected based on pilot experiments to ensure stable convergence and high-fidelity reconstructions. More details are given in the Appendix.

**LoRA Implementation:** We build on the standard LoRA formulation described in Section 4.3. For all experiments, the adapter rank is fixed at $r = 32$. The scaling factor is set to $\alpha = 1.0$ for all characteristics, except for subtlety, where we adopt a variable scaling $\alpha(s) = 2^{2+s}$, with $s$ denoting the annotated subtlety levels (1–5). The down-projection matrix is initialized with Kaiming uniform initialization, while the up-projection matrix is initialized with zeros.

**LoRA Integration with DiT Architecture :** LoRA adapters are inserted into the DiT-XL/2 attention mechanism, with Query–Key–Value (QKV) projections as the primary adaptation targets and output projections as secondary targets. This design allows efficient characteristic-specific adaptation while keeping the 675M-parameter backbone frozen. The total LoRA parameters per characteristic amount to approximately 6.2M parameters, representing only 0.9% of the base model's parameters.

## B.1. Hyperparameters

$$\mathcal{L}_{\text{inpaint}} = \mathbb{E}_{x,m,\epsilon,t} \left[ \left\| \epsilon - \epsilon_\theta \left( \sqrt{\bar{\alpha}_t}(x \odot (1 - m)) + \sqrt{1 - \bar{\alpha}_t}\,\epsilon, m, t \right) \right\|^2 \right] \tag{5}$$

$$\mathcal{L}_{\text{contrastive}} = -\sum_i \log \frac{\exp\left(\text{sim}(z_i, z_i^+)/\tau\right)}{\sum_j \exp\left(\text{sim}(z_i, z_j^-)/\tau\right)} \tag{6}$$

$$\mathcal{L}_{\text{ortho}} = \left\| W_a^\top W_b - I \right\|_F^2 \tag{7}$$

Table 6: Backbone Training Hyperparameters (DiT-XL/2)

| Parameter | Value |
|---|---|
| Model type | DiT-XL/2 |
| Input size (latent) | 32 |
| Patch size | 2 |
| Hidden size | 1152 |
| Depth | 28 |
| Attention heads | 16 |
| MLP ratio | 4.0 |
| Epochs | 50,000 |
| Batch size | 80 |
| Learning rate | $1 \times 10^{-4}$ |
| Optimizer | AdamW ($\beta_1 = 0.9$, $\beta_2 = 0.999$, $\epsilon = 10^{-8}$) |
| Noise schedule | Linear, $T = 1000$ steps |
| Context conditioning | Concat-transformer with SpatialConv+Drop |
| CFG scale | 4.0 |

Table 7: LoRA Adapter Training Hyperparameters

| Parameter | Value |
|---|---|
| Rank ($r$) | 32 |
| Scaling $\alpha$ | 1.0 |
| Training method | noxattn |
| Epochs | 150 |
| Batch size | 200 |
| Learning rate | $5 \times 10^{-5}$ |
| Optimizer | AdamW |
| Scheduler | Constant |
| Weight decay | 0.01 |
| Precision | FP32 |

Table 10: LoRA Parameter Distribution in DiT-XL/2

| Component | Dimensions | Per Block | Total |
|---|---|---|---|
| QKV Adapters | $32 \times 1152 + 3456 \times 32$ | 147,456 | 4,128,768 |
| Proj Adapters | $32 \times 1152 + 1152 \times 32$ | 73,728 | 2,064,384 |
| **Total per Adapter** | – | 221,184 | **6,193,152** |

## Appendix C. Characteristic Definitions

**Homogeneity:** Homogeneity refers to the uniformity of radiographic density (intensity levels) within a pulmonary nodule throughout its entire cross–sectional area. In contrast,

Table 8: Contrastive Fine-Tuning Hyperparameters

| Parameter | Value |
|---|---|
| Temperature $\tau$ | 0.07 |
| Margin | 1.0 |
| Feature dimension | 1152 |
| Pooling | Mean pooling |
| Feature normalization | L2 norm |
| Positive pairs | Subtlety-based grouping |
| Negative pairs | Random shuffle |
| Min pos/neg samples | 2 each per batch |

Table 9: Orthogonality Regularization Hyperparameters

| Parameter | Value |
|---|---|
| Ortho weight | 0.5 |
| Loss type | Frobenius norm |
| Target | Identity matrix |
| Adapter pairs | (calcified, homogeneous), (irregular, homogeneous), (calcified, irregular) |
| Training strategy | Alternating batches + joint optimization |
| Gradient accumulation | 2 |

non-homogeneous (heterogeneous) nodules show uneven density patterns, with some areas appearing brighter and others darker, often indicative of malignancy.

**Boundary morphology (regular vs. irregular):** Regular nodules have smooth, well–defined borders with clear demarcation from lung tissue. Irregular nodules show variable characteristics, including spiculated edges, lobulated contours, or poorly defined margins that blend with surrounding tissue, commonly associated with malignancy.

**Calcification:** Calcified nodules are characterized by high radiographic intensity and are generally smaller in size. Calcification, resulting from calcium deposits, is often associated with benign nodules and appears brighter than the surrounding tissue.

**Subtlety:** Subtle nodules refer to pulmonary lesions that demonstrate minimal radiographic contrast with surrounding lung parenchyma, making them challenging to detect on standard chest X-ray imaging. These nodules typically exhibit low-density characteristics with opacity levels that closely approximate normal lung tissue, resulting in poor visual conspicuity against the background. From the subtlety distribution analysis of our annotated scores, nodules cover a wide spectrum, with most having low subtlety scores (more subtle) and fewer having high scores (more visible).

**Nodule size:** Nodule size represents a critical malignancy risk factor, with larger nodules generally indicating higher malignancy probability. However, characteristics typically manifest in combination rather than isolation. Benign nodules commonly present as homogeneous lesions with regular margins and calcification, while malignant nodules frequently exhibit heterogeneous texture with ill-defined borders. The inherent difficulty of detecting subtle nodules underscores the importance of synthetic data generation that incorporates multiple co-occurring characteristics for improved detection and diagnosis models.

## Appendix D. Radiologist Evaluation Protocol

### Task 1: Real vs. Synthetic Nodule Assessment

**Background:** To assess the visual realism of synthetic nodules, we inserted AI-generated nodules into authentic chest X-rays and asked radiologists to distinguish them from real clinical nodules.

    **Data:** The evaluation set comprised **50 chest X-ray images** containing nodules:

- **Real nodules:** Pathological findings from patient scans.

- **Synthetic nodules:** AI-generated nodules blended into authentic radiographs.

    **Procedure:** Radiologists reviewed each image and gave a binary response:

- **Yes (Real):** Nodule appears clinically genuine.

- **No (Synthetic):** Nodule appears AI-generated.

    **Goal:** This task measured how convincing AI-generated nodules appear relative to real clinical nodules.

### Task 2: Characteristic Verification

**Background:** We next evaluated whether synthetic nodules accurately reflected specific radiological characteristics.

    **Data:** Five morphological categories were tested, with **10 images per characteristic**: Calcified, Homogeneous, Inhomogeneous, Irregular Border, and Regular Border. Each set of images was organized into a separate folder with an annotation sheet.

    **Procedure:** For each image, radiologists judged whether the nodule matched the stated feature:

- **Yes:** Exhibits the described characteristic.

- **No:** Does not match the characteristic.

    **Goal:** This task evaluated the morphological fidelity of AI-generated nodules across clinically relevant categories.

### Task 3: Subtlety Ranking

**Background:** Subtlety, or how easily a nodule can be perceived, is clinically important. We generated nodules at different subtlety levels using our diffusion-based framework.

    **Data:** Radiologists received **20 sets of images**, each containing **3 versions of the same nodule** rendered at increasing levels of subtlety.

    **Procedure:** Within each set, radiologists ranked the three images:

- Lowest Subtlety (hardest to detect) $\rightarrow$ Highest Subtlety (easiest to detect).

    **Goal:** This task tested whether the generative model produced nodules with perceptible and clinically meaningful differences in subtlety.

**Summary**

Together, these tasks: (1) Real vs. Synthetic classification, (2) Characteristic verification, and (3) Subtlety ranking, provided a comprehensive evaluation of realism, morphological fidelity, and perceptual detectability. This structured protocol ensured rigorous clinical validation of AI-generated nodules.

## Appendix E. Results And Analysis

### E.1. Subtlety LoRA Evaluation

We generated subtle nodules with Subtlety LoRA($\alpha < 24$) and assessed their impact on classification performance using the JSRT dataset, which provides a 5-level subtlety grading. As shown in Table 11, at the highest subtlety level (S1), sensitivity increased by 12% at Youden index threshold.

Table 11: JSRT classification accuracy across subtlety levels (S1 = most subtle , S5 = least subtle ).

| Train | S5 | S4 | S3 | S2 | S1 |
|---|---|---|---|---|---|
| 10k Real | 100% | 96% | 70% | 69% | 40% |
| 10k Real + 12k Fake | 100% | 100% | 76% | 76% | 52% |

### E.2. Comparison with Existing Methods

Table 12 reports quantitative scores for both full-patch synthesis and masked-patch inpainting. Across all metrics, our method consistently outperforms the GAN and fill-based methods, demonstrating its superiority in synthesizing realistic lung nodules.

Table 12: Comparison of generation methods on *ChestX-ray14*

| Method | Full | | | Masked | | |
|---|---|---|---|---|---|---|
| | PSNR | SSIM | FID | PSNR | SSIM | FID |
| ACGAN | 37.18 | 0.916 | 0.534 | 26.51 | 0.768 | 0.831 |
| ReACGAN | 37.44 | 0.916 | 0.604 | 27.54 | 0.786 | 1.227 |
| CR-Fill | 37.93 | 0.918 | 0.522 | 29.50 | 0.830 | 0.781 |
| DiT-XL/2 (Ours) | **38.74** | **0.920** | **0.390** | **34.26** | **0.892** | **0.475** |

### E.3. Comparison of CFG Control versus Separate LoRA for Label Guidance

Using classifier-free guidance (CFG) to control both nodule characteristics and the mask leads to suboptimal adherence to characteristic-specific attributes. To evaluate this, we perform an ablation comparing models trained with CFG-based label control against our approach using separate LoRA adapters. We compute FID scores for each characteristic, and as shown in Table 13, LoRA-based control yields consistently lower FID values, indicating stronger characteristic fidelity.

Table 13: FID comparison between CFG-based label control and LoRA-based control across nodule characteristics.

| Characteristic | CFG Control (FID) | LoRA Control (FID) |
|---|---|---|
| Calcification | $15.87 \pm 1.51$ | $2.29 \pm 0.09$ |
| Regular Border | $4.04 \pm 0.84$ | $1.96 \pm 0.12$ |
| Irregular Border | $4.19 \pm 0.59$ | $2.90 \pm 0.03$ |
| Homogeneous Texture | $8.48 \pm 1.50$ | $5.71 \pm 0.36$ |
| Inhomogeneous Texture | $12.58 \pm 1.94$ | $8.13 \pm 0.28$ |

### E.4. Radiologist evaluation with confidence intervals

**Pooled vs. majority-vote metrics.** We report results using two aggregation schemes across radiologists. *Pooled (vote-level)* metrics treat each radiologist response as one independent vote. If there are $N$ cases (images or ranking sets) and $R$ radiologists, the pooled denominator is $n = N \times R$. The pooled rate answers: "Across all individual ratings, how often was the target response selected?" *Majority-vote (case-level)* metrics first aggregate the $R$ votes per case into a single panel decision (e.g., $\geq 2/3$ agreement), and then compute performance across cases with denominator $n = N$. The majority-vote rate answers: "For how many cases did the panel agree with the target outcome?" This provides an image level measure of consensus. We also provide 95% confidence intervals ( Wilson Score )

**Task 1: Real vs. synthetic** Radiologists were presented with real and synthetic samples and asked to judge whether each sample appears *real* or *synthetic*. We report (i) the fraction of real images judged as real and (ii) the fraction of synthetic images judged as real, using both pooled and majority-vote aggregation. The results are shown in table 14.

**Task 2: Characteristic verification** Radiologists were shown generated nodules targeting a specific radiological characteristic (e.g., border type, homogeneity, calcification) and asked whether the target characteristic is present. We report pooled and majority-vote "Yes" rates per characteristic to quantify how reliably the intended attribute is recognized by experts. The results are shown in table 15.

**Task 3: Subtlety ordering** Radiologists were presented with small sets generated at different subtlety levels and asked to rank/order them by subtlety. Each set is scored as *correct* vs. *incorrect* ordering, and we report pooled and majority vote ranking accuracy, measuring how consistently the intended subtlety control aligns with expert perception. The results are shown in table 16.

Table 14: Task 1: Real vs Synthetic summary with 95% CI

| Metric | Real images | | Synthetic images | |
|---|---|---|---|---|
| | **Rate** | **95% CI** | **Rate** | **95% CI** |
| Pooled "Looks real" rate | 0.867 | $[0.703, 0.947]$ | 0.700 | $[0.521, 0.833]$ |
| Majority-vote "Looks real" rate | 0.900 | $[0.596, 0.982]$ | 0.800 | $[0.490, 0.943]$ |

Table 15: Task 2: Characteristic verification Pooled + Majority-vote with 95% CI

| Characteristic | Pooled "Yes" rate | | Majority-vote "Yes" rate | |
|---|---|---|---|---|
| | Rate (k/n) | 95% CI | Rate (k/n) | 95% CI |
| Homogeneous | $24/30 = 0.800$ | $[0.627, 0.905]$ | $9/10 = 0.900$ | $[0.596, 0.982]$ |
| Inhomogeneous | $22/27 = 0.815$ | $[0.633, 0.918]$ | $9/9 = 1.000$ | $[0.701, 1.000]$ |
| Irregular border | $27/30 = 0.900$ | $[0.744, 0.965]$ | $10/10 = 1.000$ | $[0.722, 1.000]$ |
| Regular border | $28/30 = 0.933$ | $[0.787, 0.982]$ | $9/10 = 0.900$ | $[0.596, 0.982]$ |
| Calcified | $20/30 = 0.66$ | $[0.488, 0.808]$ | $8/10 = 0.800$ | $[0.490, 0.943]$ |

Table 16: Task 3 (Subtlety ranking): Accuracy with 95% CI

| Metric | Result (k/n) | Accuracy | 95% CI |
|---|---|---|---|
| Pooled | $20/30$ | $0.667$ | $[0.488, 0.808]$ |
| Majority-vote | $8/10$ | $0.800$ | $[0.490, 0.943]$ |

### E.5. Orthogonality Verification via Adapter Weight Analysis

To verify that the proposed orthogonality loss increases subspace orthogonality rather than trivially shrinking adapter weights, we analyze the behavior of adapter weight matrices under different training configurations. In particular, we compare adapters trained separately against configurations where two adapters are trained jointly with Frobenius-norm regularization.

Table 17 reports the aggregate magnitude of the adapter weight matrices across multiple characteristic pairings. Despite the presence of Frobenius-norm regularization, the jointly trained adapters consistently exhibit higher weight magnitudes compared to separately trained counterparts.

Table 17: Comparison of adapter weight magnitudes for separately trained adapters versus two adapters trained jointly with Frobenius-norm regularization.

| Comparison | Separate | Two Adapters (Frobenius) | % Change |
|---|---|---|---|
| Calcified vs. Homogeneous | 329.01 | 430.89 | $+30.97\%$ |
| Homogeneous vs. Calcified | 388.23 | 502.42 | $+29.41\%$ |
| Irregular vs. Homogeneous | 539.28 | 711.84 | $+32.00\%$ |
| Homogeneous vs. Irregular | 388.23 | 803.59 | $+106.99\%$ |

Notably, this increase in weight magnitude occurs even though earlier results show a reduction in the Frobenius norm of individual adapters. This indicates that the orthogonality constraint does not simply suppress adapter activations. Instead, it encourages adapters to

occupy more distinct directions in parameter space, leading to improved subspace separation.

These findings support the claim that the proposed orthogonality loss promotes genuine representational disentanglement between adapters, rather than acting as a magnitude-reducing regularizer.

## E.6. Computational Cost Analysis

This section summarizes the computational requirements of the proposed method, including training time, inference time, and hardware usage.

All experiments were conducted on NVIDIA L40 GPUs (48 GB memory). Backbone training was performed in two stages. In Stage 1, the DiT backbone was trained for approximately one week using four L40 GPUs. In Stage 2, training was continued for approximately two additional days on four L40 GPUs. After Stage 2, the backbone parameters were frozen for all subsequent experiments.

The DiT-XL/2 backbone contains approximately 682M parameters. During adapter training, we optimized only the LoRA parameters, with each adapter comprising 6.2M parameters (approximately 0.91% of the backbone).Each LoRA adapter was trained for 150 epochs with a batch size of 25 on $256 \times 256$ nodule-centered patches. Training a single adapter required approximately 10 hours on four L40 GPUs. Jointly training two adapters with the proposed orthogonality constraint required approximately 20-30 hours on four L40 GPUs, depending on the characteristic pairing and dataset size.

Inference was performed with 250 DDIM sampling steps, with an average runtime of approximately 10 seconds per sample. Since the backbone remains frozen at inference time, the computational cost is dominated by diffusion sampling rather than adapter-specific operations.

Overall, the proposed approach remains computationally practical, providing parameter-efficient adaptation and manageable inference cost relative to full backbone fine-tuning.

## E.7. Extending Orthogonality Loss beyond two characteristics

The orthogonality-based merging objective is not restricted to two attributes, it generalizes directly to K adapters by enforcing orthogonality across all selected LoRA updates (e.g., pairwise across the set), so in principle the same framework can synthesize nodules with three or more characteristics simultaneously.

$$\mathcal{L}_{\text{orth}} = \sum_{i \neq j} \|W_i^\top W_j\|_F^2 \tag{8}$$

We did not include more than 2 characteristic compositions or downstream augmentation experiments in this submission for two practical reasons. First, our available datasets do not contain sufficiently dense co-occurrence annotations (i.e., reliable labels for multiple attributes on the same nodule) to form training/evaluation splits that would support a fair quantitative study of higher-order compositions. Second, as the number of characteristics increases, many combinations become rare or clinically incompatible in practice, leading to very small sample sizes and unstable estimates for both synthesis evaluation and downstream detection benchmarking.

**E.8. Detection gains plateauing with more synthetic data:**

We can think of two reasons why this is the case:

- Even if images look realistic, large batches of synthetic samples can be less diverse than real (mode concentration around common textures/borders), so additional samples add less new information than expected, leading to saturation.

- After some point, adding more synthetic nodules can over-emphasize the synthetic distribution relative to the real one, so the detector starts fitting synthetic-specific cues instead of generalizable ones.

