# OpenReview forum: "A Diffusion-Driven Fine-Grained Nodule Synthesis Framework for Enhanced Lung Nodule Detection from Chest Radiographs"
_MIDL.io/2026/Conference — MIDL 2026 Poster_

### Official Review · Reviewer_zn8i · 2026-01-10

**Confidence:** 5
**Preliminary Rating:** 4
**Final Rating:** 5

**Summary:**

This paper proposes a diffusion-based framework for fine-grained pulmonary nodule synthesis in chest radiographs, aiming to address data scarcity and limited diversity in training datasets for lung nodule detection. The method builds on a Diffusion Transformer backbone with mask-conditioned nodule placement and introduces characteristic-specific LoRA adapters to control radiologically relevant attributes such as calcification, border regularity, texture homogeneity, and subtlety. To enable multi-attribute synthesis, the authors propose an orthogonality-constrained LoRA merging strategy that reduces interference between adapters. The approach is evaluated through image quality metrics, radiologist studies assessing realism and controllability, and downstream nodule detection experiments on both in-house and public datasets, demonstrating consistent performance improvements when augmenting real data with generated nodules.

**Strengths:**

The paper addresses an important and clinically relevant problem: improving lung nodule detection in chest radiographs by generating synthetic data that captures diagnostically meaningful characteristics. The use of diffusion models, specifically a Diffusion Transformer backbone, is well motivated given their stability and high-fidelity generation compared to GAN-based approaches. A key strength is the explicit focus on controllability of multiple radiological attributes rather than generic realism, which aligns with clinical needs and should differentiate this work from prior nodule synthesis methods.

The characteristic-specific LoRA adapters are a practical and parameter-efficient design choice, allowing fine-grained control without retraining the full backbone.

The orthogonality-constrained merging strategy is valid and addresses a known limitation of naive LoRA composition, with both qualitative examples and quantitative Frobenius norm analysis supporting reduced interference.

The experimental evaluation is comprehensive, inclusing standard image quality metrics like FID, structured radiologist studies, and, importantly, downstream detection experiments that show consistent improvements in AUC and IoU across multiple datasets. The inclusion of public benchmarks strengthens the credibility of the results, and the paper is generally well organized, clearly written, and grounded in relevant prior work.

**Weaknesses:**

- Evaluation metrics focus on patch-level image quality (all PSNR, SSIM, FID) that do not directly assess anatomical plausibility or structural consistency within the broader chest context. Although downstream detection performance partially addresses this, the analysis does not disentangle whether gains stem from improved realism, better attribute control, or simply increased data volume. Additionally, improvements plateau or slightly degrade at higher levels of synthetic augmentation, but this behavior is not deeply analyzed.


- The radiologist evaluation, while valuable, is a bit limited in scale, involving only three experts and a relatively small number of samples. Though it is understandalbe that including radiologist in this sort of work can be hard. The reported agreement percentages lack confidence intervals or inter-reader agreement statistics, making it difficult to assess robustness in the set low number of participants.

- Several methodological choices would benefit from clearer justification or stronger ablation. The selection of LoRA ranks, scaling factors for subtlety, and the orthogonality loss weight appear empirically chosen, but their sensitivity is not explored - the paper would benefit from some comments about these issues if they were explored. The multi-attribute evaluation is restricted to two-characteristic combinations, leaving open questions about scalability to more complex attribute mixtures.

- The framework relies on large in-house datasets and extensive pretraining, which may limit accessibility and reproducibility for smaller research groups.

**Detailed Comments:**

The paper would benefit from clearer separation between contributions arising from the diffusion backbone and those attributable specifically to the proposed LoRA and merging strategy. A sensitivity analysis on the orthogonality regularization weight and LoRA scaling parameters would strengthen the methodological section. Including structure-aware evaluation metrics or qualitative failure cases could further improve validation. Some figures are could benefit from more explicit guidance on what differences the reader should focus on. Overall presentation quality is good, with only minor opportunities for improved clarity.

**Justification Of Final Rating:**

After considering the authors’ discussion and updated manuscript, the concerns raised in the initial review have been adequately addressed. The rebuttal provides explanations for the choice of orthogonality regularization strength and LoRA rank, including a discussion of stability regimes and trade-offs, which improves confidence in the robustness of the proposed method even in the absence of additional ablations.  The authors also appropriately acknowledge the limitation that downstream performance gains cannot yet be fully disentangled between attribute-specific control and generic data augmentation, and they clearly outline the experimental design required to study this in future work. The text has been updated with radiologist study statistics and scalability to more characteristics.  Overall, while some limitations remain, the rebuttal satisfactorily resolves the main points of clarification.

**Justification Of The Preliminary Rating:**

Overall, this paper makes a well-substantiated contribution to data augmentation for lung nodule detection. It addresses a clinically important problem which is data scarcity and limited diversity in chest radiographs., using a diffusion-based framework that is designed with clinical interpretability in mind. The integration of a mask-conditioned Diffusion Transformer with characteristic-specific LoRA adapters is novel in this specific context, and the proposed orthogonality-constrained LoRA merging strategy meaningfully addresses known challenges in multi-adapter composition, supported by both theoretical motivation and empirical evidence. The experimental validation is thorough, spanning image quality metrics, radiologist evaluations, and downstream detection performance on both in-house and public datasets, with consistent and practically relevant improvements. Although additional sensitivity analyses and larger-scale reader studies could further strengthen the work, these limitations are relatively minor and do not detract from the overall technical quality, clarity, and potential impact of the paper, justifying an accept recommendation.

**Questions To Address In The Rebuttal:**

How sensitive is the performance to the choice of orthogonality regularization strength and LoRA rank, and are there regimes where merging becomes unstable?

To what extent do downstream detection improvements arise from attribute-specific control versus generic data augmentation effects?

Can the authors comment on how the method would scale to combining more than two characteristics simultaneously?

Can additional statistical analysis or confidence measures be provided for the radiologist evaluation to strengthen its interpretability?

---

> ### Author Response · Authors · 2026-01-25
>
> ### **Questions Addressed In The Rebuttal:**
>
> 1.**Sensitivity/instability: How do LoRA rank and orthogonality weight affect merging stability/performance?**
>
> A loss weight of 0.5 was used during training for orthogonal training. This value was chosen empirically to ensure that the regularization term remained of the same order of magnitude as the diffusion loss. There can be intuitive regimes where merging can become *effectively* unstable:
> - λ too low (weak orthogonality): independently trained LoRAs tend to be non-orthogonal, and since all attributes operate on the *same nodule region*, their updates interfere. This is exactly when we empirically see issues like one characteristic dominating and artifacts near mask boundaries with naive merging
> -  λ too high (over-regularized): the adapters are forced to be “too separated,” which can reduce how strongly a characteristic can be expressed (since λ trades off with reconstruction/fit)
>
> Regarding LoRA rank sensitivity, we keep the rank fixed at r = 32
> -  Lower ranks can under-express fine-grained morphology (reduced adapter capacity).
> - Higher ranks increase adapter capacity but can also increase the chance of overlap/interference unless orthogonality is enforced
>
> We look forward to ablation on these regime as a part of future work
>
> 2. **Source of gains: Are downstream improvements from attribute control or just more synthetic data?**
>
> In our setting, attribute-specific control is motivated by clinical interpretability: radiological traits such as calcification, border definition, and homogeneity (along with size, location, and subtlety) are relevant cues for suspicious nodules on CXR (e.g., larger, ill-defined, non-calcified, inhomogeneous nodules are more concerning and typically warrant follow-up). By explicitly controlling these attributes, we can synthesize harder and clinically meaningful cases which can be suspicious for lung cancer.
> With the current experiments, however, we cannot precisely disentangle how much of the downstream gain comes from generic augmentation (more positive samples) versus targeted diversity via attribute control. A clean decomposition would require a controlled study at a fixed synthetic budget, comparing *random/uncontrolled* synthetic augmentation against *attribute-targeted* augmentation with matched sample counts. We leave this as a part of future work.
>
> 3. **Can the method scale to composing three+ attributes?**
>
> The regularization term naturally extends to scenarios involving more than two adapters its shown in Appendix section E7 ( eq. 8).   We did not evaluate >2-way compositions in this work because our datasets do not provide sufficiently dense multi-attribute co-annotations to reliably train and quantitatively validate higher-order combinations. Moreover, as the number of characteristics increases, many of our chosen characteristics combinations become rare or effectively mutually exclusive in clinical data.
>
> 4. **Radiologist stats: Can you add confidence/uncertainty measures for the reader study?**
>
> A detailed results with confidence intervals have been added in Appendix section E4.

---

> > ### Comment · Reviewer_zn8i · 2026-01-31
> >
> > Thank you for the comments. Overall, the responses address the core concerns raised in the review and clarify several methodological choices.
> >
> > For orthogonality regularization, seems reasonable, though explicit ablation results would have strengthened it, the explanation is sufficient for the current scope. Same for clarification on LoRA rank.
> >
> > The response on the source of downstream performance gains is clear and appropriately honest. They acknowledge that the current experiments cannot separate the effects of attribute-specific control from those of generic data augmentation. While this remains a limitation, it does not detract from the paper’s main contribution.
> >
> > I think for the possible scope this seems sufficient for the accept, future work can be done as a follow-up paper on controlled augmentation studies and other detailed points which can be a  valuable extension.

---

### Official Review · Reviewer_yhBv · 2026-01-14

**Confidence:** 3
**Preliminary Rating:** 2
**Final Rating:** 3

**Summary:**

This paper proposes a diffusion-based framework for synthesizing lung nodules on chest radiographs (CXRs) with controllable radiological characteristics. The authors build upon DiT-XL/2 as the backbone, conditioning on binary masks for spatial control, and introduce characteristic-specific LoRA adapters for attributes including calcification, border regularity, homogeneity, and subtlety. The key technical contribution is an orthogonality-constrained LoRA merging strategy using a Frobenius norm penalty $\mathcal{L}_{orth} = \|W_1 W_2^\top\|_F^2$ to reduce interference when combining multiple adapters for multi-characteristic synthesis. The framework is trained on a large in-house dataset (1.2M CXRs) and evaluated through radiologist assessments, standard image quality metrics (FID, PSNR, SSIM), and downstream nodule detection performance on JSRT and ChestX-ray14, demonstrating AUC improvements from 0.856→0.902 and 0.901→0.932 respectively when augmenting training data with synthetic nodules.

**Strengths:**

1. **Clinically Motivated Problem**: The paper addresses a genuine challenge in CAD development, i.e. the scarcity of annotated CXRs with diverse nodule characteristics. The focus on radiologically meaningful attributes (calcification, border definition, homogeneity, subtlety) is well-aligned with clinical practice.

2. **Systematic Framework Design**: The three-stage training pipeline (base pre-training → mask-conditioned fine-tuning → characteristic-specific LoRA training) is logical and modular, allowing efficient adaptation without full model rethinking.

3. **Novel Orthogonality Constraint**: The identification of non-orthogonal parameter spaces as a source of interference during LoRA composition is insightful. Figure 6 provides evidence that cross-training reduces layer-wise Frobenius norms compared to separately trained adapters.

4. **Comprehensive Evaluation Protocol**: The authors conduct multi-faceted evaluation including radiologist assessment (realism, controllability, subtlety ranking), quantitative metrics, and downstream task performance across multiple datasets.

5. **Subtlety Slider**: Adapting Concept Sliders with $\alpha(s) = 2^{2+s}$ for graded subtlety control addresses an often-overlooked aspect of nodule detectability.

6. **Demonstrated Downstream Utility**: Consistent improvements in nodule detection AUC across both in-house and public test sets suggest practical value for CAD augmentation.

**Weaknesses:**

1. **Orthogonality Loss Formulation**: The proposed $\mathcal{L}_{orth} = \|W_1 W_2^\top\|_F^2$ doesn't enforce true subspace orthogonality between LoRA adapters. This loss can be minimized by simply reducing adapter magnitudes, confounding orthogonality with expressiveness. A normalized formulation $\|\hat{W}_1 \hat{W}_2^\top\|_F^2$ would be more principled.

2. **No Quantitative Ablation for Orthogonality Loss**: Despite $\mathcal{L_orth}$ being presented as a key contribution, there is no quantitative evaluation demonstrating that including this loss improves downstream performance. Figure 6 shows Frobenius norms decrease with cross-training, but this only verifies the loss functions as designed—not that it helps detection. Figure 7 provides only qualitative visual comparisons. The logical chain "lower Frobenius norm → better merging → better synthesis → better detection" is assumed but never validated. A critical ablation comparing merged LoRAs with vs. without $\mathcal{L}_{orth}$ on downstream AUC/IoU is missing.

3. **Weak Baseline Comparisons**: Comparisons are limited to older methods (ACGAN 2017, ReACGAN 2021, CR-Fill 2021). Contemporary diffusion-based medical image synthesis methods (RoentGen, medical ControlNet variants) are not compared, making it unclear whether improvements stem from the proposed contributions or simply from using diffusion models.

4. **Statistical Inadequacies**: No confidence intervals, standard deviations, or significance tests are reported. The non-monotonic AUC progression in Table 3 (drops at 4k, 8k synthetic samples) suggests high variance. Furthermore, Table 3 and Table 5 report inconsistent AUC values for apparently identical experiments on ChestX-ray14 (0.9318 vs 0.9518).

5. **Critical Missing Ablations**: Beyond the orthogonality loss, no ablations on LoRA rank $r$, orthogonality weight $\lambda$, CFG scale, or VAE fine-tuning are provided. Only pairwise characteristic combinations are evaluated despite claims of extensibility.

6. **Data Protocol Concerns**: The segmentation model used to refine public dataset masks was trained on in-house data, introducing potential bias. The exact train/test splitting protocol and characteristic distribution across splits are unclear.

**Detailed Comments:**

- **Equation 3** uses $\alpha(s) = 2^{2+s}$ giving scaling factors of 8-128, which is unusual compared to standard LoRA scaling of $\alpha/r$. Please clarify the rationale.

- The "Short Title" placeholder appears throughout the document, suggesting incomplete preparation.

- **Table 1** shows different nodule prevalence rates across splits (2.5% in diffusion trainset, 10% in downstream trainset, 16.7% in test set). How does this distribution shift affect results?

- The downstream evaluation (Tables 3-5) only evaluates the base diffusion model and individual characteristic LoRAs. There is no downstream evaluation of the merged LoRA outputs, which is the setting where $\mathcal{L}_{orth}$ would matter most.

- Consider analyzing which specific nodule types benefit most from synthetic augmentation (subtle vs. obvious, small vs. large).

- Missing computational cost analysis (training time, inference time, GPU requirements).

- Consider comparing with multi-characteristic training in a single LoRA versus separate LoRAs to justify the proposed approach.

**Justification Of Final Rating:**

I thank the authors for addressing some of my concerns, and I am happy to see the newly included Table 17, which is very convincing w.r.t. LoRA orthogonality. Nevertheless, I would argue that an ablation on the orthogonality loss regarding downstream performance would strengthen the paper substantially, as downstream performance is the eventual goal. Since the authors clearly have access to the trained adapter weights (as evidenced by Table 17's magnitude analysis), running a small-scale downstream experiment comparing merged LoRAs with vs. without $\mathcal{L}_{orth}$ should be readily feasible.

For now I have updated my score to borderline, and am willing to increase it to weak accept given the inclusion of this experiment (or at least a discussion thereof)

**Justification Of The Preliminary Rating:**

While this paper addresses a clinically relevant problem and presents a reasonable framework for controllable nodule synthesis, several fundamental issues prevent acceptance. Most critically, the orthogonality loss $\mathcal{L_orth}$, which is presented as a key contribution, lacks any quantitative validation that it improves downstream detection performance. The only evidence provided is that Frobenius norms decrease (Figure 6) and qualitative visual comparisons (Figure 7), but no ablation shows that removing $\mathcal{L}_{orth}$ hurts AUC/IoU on the detection task. This makes it difficult to assess whether this contribution actually matters.

The orthogonality loss formulation itself has theoretical issues (confounded with magnitude) and lacks rigorous validation that it actually improves subspace separation rather than simply reducing weights. The experimental evaluation has significant gaps: no statistical significance testing despite high-variance results, comparisons only with non-diffusion baselines, and unexplained inconsistencies between tables.

The contributions, while sensible from an engineering perspective, are incremental combinations of existing techniques (DiT, LoRA, mask conditioning, Frobenius orthogonality) without sufficient validation of the claimed improvements. Addressing the orthogonality ablation, statistical rigor, and baseline comparisons would substantially strengthen this work.

**Questions To Address In The Rebuttal:**

1. **Orthogonality Ablation**: Can you provide a quantitative comparison of downstream detection performance (AUC/IoU) when training merged LoRAs *with* vs. *without* $\mathcal{L}_{orth}$? This is essential to validate the claimed contribution. Currently, Figure 6 only shows that Frobenius norms decrease, not that this improves the actual downstream task.

2. **Orthogonality Weight Sensitivity**: What is the effect of varying $\lambda$ on both the Frobenius norm and downstream performance? Is there a trade-off between orthogonality and individual characteristic fidelity?

3. **VAE Reconstruction**: Can you provide quantitative analysis of VAE reconstruction quality specifically within nodule regions, stratified by characteristic? Have you compared with a CXR-fine-tuned VAE?

4. **Orthogonality Verification**: Can you demonstrate that the orthogonality loss actually increases subspace orthogonality (e.g., principal angle analysis) rather than simply reducing adapter magnitudes?

5. **Table Inconsistency**: Why does "10k real + 10k syn" yield 0.9318 AUC in Table 3 but 0.9518 AUC in Table 5 for ChestX-ray14?

6. **Contemporary Baselines**: Can you compare with recent diffusion-based medical image synthesis methods?

7. **Three+ Characteristics**: Can you demonstrate synthesis with more than two characteristics simultaneously, and evaluate downstream performance with such augmentation?

---

> ### Author Response · Authors · 2026-01-25
>
> ### **Detailed Comments Addressed**
> 1. **Eq. 3 LoRA scale justification:**   For subtlety control in Eq. (3), we empirically found that using an exponential scaling of the LoRA factor yields more stable and monotonic control than a linear scale. With linear scaling, small α values resulted in weak updates, where the base diffusion model’s default nodule appearance dominated the LoRA modification. Exponential scaling increases the effective separation between subtlety levels, ensuring the LoRA update meaningfully influences the generation even at low conspicuousness, while maintaining smooth progression across grades. This design choice is empirically motivated and validated through radiologist ranking experiments, and we note that alternative scaling functions remain an open direction for future work.
>
> 2. **Document formatting(Short Title):** - We will be correcting this in the revised submission.
> 3.  **Missing computational cost analysis:** - We have provided detailed computational cost analysis in Appendix Section E6
> 4. **Comparison between a single multi-characteristic LoRA and separate per-characteristic LoRAs:** - We did explore training a single LoRA jointly on multiple characteristics, but on visual inspection  it provided weaker and less disentangled control than using separate characteristic-specific LoRAs.
>
> Why this happens (insight):
>
> - The targeted attributes (e.g., border definition, homogeneity, calcification, subtlety) are expressed in the same localized region (the nodule), so the gradients for different attributes often overlap and compete. A single adapter tends to learn a coupled representation that correlates attributes instead of keeping them separable.
> - Several characteristics are also statistically correlated in real data (and some combinations are rare), so joint training can collapse toward the most common co-occurrence patterns rather than learning clean, independent “axes” of control.
>
> ### **Questions Addressed In The Rebuttal:**
> 1. **Orthogonality ablation: downstream AUC/IoU:** -  We haven't performed this  ablation and would like to take this up in future work.
>  2. **λ sensitivity: effect of varying orthogonality weight on norm + downstream; trade-off with attribute fidelity :** - Our method preserves the ability to independently control the strength of each characteristic (i.e., each LoRA adapter) at inference time via dedicated scaling hyperparameters. This level of control is not supported by methods such as ZipLoRA, where adapter contributions are implicitly merged. By tuning these hyperparameters, users can explicitly regulate how strongly each characteristic is expressed, while the orthogonality constraint ensures that adapters remain well-separated and combine coherently without interfering with one another.
> 3. **VAE quality:** - While we did not perform a standalone quantitative evaluation of VAE reconstruction quality within nodule regions, we did analyze the quality of generated images in masked nodule regions using standard metrics (PSNR, SSIM, FID), as reported in Table 12 (Appendix E.2). These metrics reflect the combined effect of the VAE and diffusion model on nodule reconstruction and synthesis. In exploratory experiments, a CXR-fine-tuned VAE did not yield noticeable improvements in these region-level generation metrics or visual quality. Consistent with prior diffusion-based CXR works, we therefore use an off-the-shelf VAE and leave isolated, region-specific VAE reconstruction analysis as future work.
> 4. **Verify orthogonality:** -  A detailed analysis has been done on this in Appendix E5. The table 17 shows an increase in the magnitude of the adapter weight matrices when trained with the orthogonality loss. Notably, this occurs even though the Frobenius norm decreases, as reported in earlier results. This behavior suggests that the orthogonality constraint encourages larger individual weight magnitudes in order to preserve distinctiveness between adapters.
> 5. **Table mismatch: ChestX-ray14 AUC inconsistency** - We acknowledge the inconsistency. The correct ChestX-ray14 AUC for “10k real + 10k syn” is 0.9318(as reported in Table 3). The value 0.9518 in Table 5 is a typographical error. We have reported the correct value (0.9318) in the text immediately above Table 5, has been corrected in the revised submission
> 6. **Contemporary Baselines comparisons:**  - To the best of our knowledge, no prior diffusion method performs mask-faithful CXR nodule synthesis with explicit, radiologist-defined multi-characteristic control; this is our main contribution. We also tried an SDXL-style backbone, but it visually underperformed DiT-XL for mask-controlled generation (weaker mask adherence/realism), and since we did not compute standardized metrics, we did not include it as a formal baseline. Our comparisons therefore focus on methods that explicitly target synthetic nodule generation.
> 7. **3+ attributes:** Discussed in detail in Appendix section E7

---

### Official Review · Reviewer_uH83 · 2026-01-17

**Confidence:** 4
**Preliminary Rating:** 4
**Final Rating:** 4

**Summary:**

The authors propose a diffusion-based framework for synthetic nodule generation in chest X-Ray images (CXRs). The authors introduce characteristic control properties in their framework, which aims to help the framework learn different nodule types better. To counter other attention-level issues, the authors also propose a loss function and demonstrate the advantage of using it. Finally, results have been presented for nodule generation, and downstream tasks like segmentation and classification.

**Strengths:**

I like the strategy of utilizing diffusion models-based nodule generation for CXR, which has currently been proven as working well with GAN- and VAE-based generative methods. The paper does show a promising direction for image generation for nodule generation. In particular, I am quite happy that a) the authors benchmark with NODE21's winning method, and b) show how image generation helps in downstream tasks.

**Weaknesses:**

There are only two parts I'd like to highlight and hear back from the authors about them:

1. The motivation to choose LoRA is strictly discussed in light of previous (unrelated) work, without any backed experiments of with vs. without LoRA. In addition, there's no reflection on the influence of rank of the matrix (and other parameters) for the adapters used. I suggest the authors take a closer look at this and present more results.
2. There's some open questions on a clear picture on the data scaling angle of the presented method. To boil it down to two strategies, a) how less is good already, and b) how much is too much?

(see the "Questions To Address In The Rebuttal" section below for a detailed outline)

**Detailed Comments:**

On a higher level, I am happy about the methods developed in the paper and the data- and method-level benchmarks presented. However, it needs some work to address a) the weaknesses mentioned above, and b) demonstrate the practical (code-level) usage of the proposed method. I would be happy to see the response from the authors for the comments shared above. Thanks!

**Justification Of Final Rating:**

I have been happy with the methods developed and the manuscript presentation right from the very beginning. The authors have indeed added more detailed sections in appendix to address all reviewers' feedback and I am happy with the response of the authors to my questions. However, the motivation of open-sourcing the codebase is still missing (with no rationale presented), release of datasets and pretrained model is even more clouded (again, no response or rationale received for this), and some rebuttal comments have not been complemented in the manuscript.

I would like to stick to my rating here, although I am unhappy about the points I mentioned above.

**Justification Of The Preliminary Rating:**

I am happy with the method developed and the manuscript presentation. However, I am unhappy about two things, a) the complexity of reproducibility for methods presented in Figure 1, and b) the missing motivation of open-sourcing of the codebase, internal dataset and pretrained models. I am open to discuss the response from authors in the coming days.

PS. Convincing me shouldn't be too difficult, as I am quite happy about the overalls in general.

**Questions To Address In The Rebuttal:**

- Page 2: "After training the backbone, we attach LoRA modules (Hu et al., 2021) for four radiological attributes: calcification, border definition (regular/irregular), homogeneity, and subtlety by using characteristic-specific subsets to capture fine-grained distinctions without affecting general nodule synthesis." -> I didn't understand how LoRA modules could be specifically mapped to some specific attribute? Did the authors map the pretrained tokens to respective (biological) characteristics of nodules, or there's something more complicated going on? Or is it an assumption that LoRA learns such characteristics? I suggest the authors clarify this.

- Table 3: It seems like the performance gains are not scaling with increasing the amount of synthetic data, despite the quality of the generated nodules pretty well. This is counter-intuitive to my assumptions and the generation quality claims presented in the paper (besides for JSRT). Do the authors have any intuitions / hypotheses why this could be the case? (e.g. even the synthetic data addition results differ on a per-data-testset level)

- Page 10: "Limitations include difficulty with some out-of-distribution generations by composition of LoRAs." -> Could the authors present the following: a) demonstrate some OOD generations with the proposed method, and b) discuss the effect of using LoRA at all vs. not using any matrix decomposition, i.e. removing LoRA strategies completely. This would bring in the value of showing how LoRA-related strategies discussed have their strengths (if at all) compared to the classic full (ranked matrix) finetuning.

- Adding to the LoRA motivation, is there a motivation towards parameter efficient finetuning at all? If not, why opt for LoRA at all? (unless the LoRA merging strategies cannot be adapted to the classic attention, which I also don't see the reason why?) I suggest the authors to consider the point presented above and show the effects of with vs. without LoRA.

- Any reason to choose the LoRA rank as 32 and the scaling ratio as 1.0? I ask because there's (several but I mention one of the most popular recent) work which has demonstrated the task-specific nature of the mentioned parameters, and the corresponding advantage of making the parameter choices (https://arxiv.org/abs/2405.09673), and similar extensions to medical imaging, presented at MIDL in last years (https://proceedings.mlr.press/v250/dutt24a.html, https://arxiv.org/abs/2502.00418). Have the authors explored this? If not, I suggest the authors add a new experiment section to back their choice of the presented parameters, since major claims of the paper is built around low-rank adapters.

- Table 3 and Table 5: I have a curious question here. How well does the presented method work for segmentation and classification, when only trained on synthetic images? This is an interesting question to ask to build pretrained models and would potentially complement the data-scaling question above. I suggest the authors to share their insights and outlook on this.

- Table 5: Another curious question coming in. Have the authors explored AUC scores when all the image generation methods combine together? i.e. 10k real + ("a" ACGAN + "b" ReACGAN + "c" CRFILL + "d" DiT-XL). One could also try the previous vs. 10k real + ("a" ACGAN + "b" ReACGAN + "c" CRFILL). This again goes in the scaling direction, to understand if the performance plateaus with a few good synthetic images, or a lot of okayish generated images.

Another final comment: there's a strong emphasis on the (LoRA-based) (effective yet extremely complex) pretraining framework (ref. Figure 1) and the strengths the internal dataset brings to the pretraining step. However I couldn't find any source code, open-source dataset, or pretrained model links in the manuscript. This goes slightly away from MIDL's motivation of open-sourcing and reproducibility in science. I strongly suggest the authors to ensure that their codebase with pretrained weights are released (and ofc ensure the framework is usable, eg. by adding example scripts for image generation), and discuss if the dataset could be open-sourced too. All of this would enunciate the value of the proposed method and make it usable by others.

---

> ### Author Response · Authors · 2026-01-25
>
> ### **Questions Addressed In The Rebuttal:**
> 1. **LoRA–attribute mapping clarification**: LoRA modules are not mapped to pretrained tokens or explicitly aligned to biological concepts in our method.  Each attribute is controlled by a dedicated LoRA trained on a radiologist-labeled, characteristic-curated subset (backbone frozen). At inference, enabling a specific adapter injects a low-rank update into the DiT, biasing generation toward the corresponding radiographic appearance.
>
> 2. **Why detection gains plateau with more synthetic data**:
>    We can think of two reasons why this is the case:
>    - Even if images look realistic, large batches of synthetic samples can be less diverse than real (mode concentration around common textures/borders), so additional samples add less new information than expected, leading to saturation.
>    - After some point, adding more synthetic nodules can over-emphasize the synthetic distribution relative to the real one, so the detector starts fitting synthetic-specific cues instead of generalizable ones.
>
> 3. **“LoRA vs no-LoRA/full finetuning” motivation** :
> We started our work by trying both mask control and label control with CFG( classifier free guidance), with full finetuning,  but we found suboptimal nodule characteristics control with it.  We think this is because CFG across multiple semantic labels competes with mask guidance and weakens boundary fidelity, especially when mask conditioning also relies on CFG. We therefore reserve CFG for the mask and handle semantic control via characteristic-specific LoRA adapters.  We have presented the results in Appendix Section E3, where we compare CFG based full finetuning control vs Separate LoRA control.
>
> 4. **Motivation for LoRA beyond parameter efficiency (why LoRA vs classic finetuning; CFG vs LoRA control)**:
> We adopt LoRA not only for parameter efficiency, but to decouple characteristic control from mask-guidance: CFG for both masks and labels degrades boundary fidelity as well as characteristic control, whereas separate LoRA adapters provide lightweight, modular attribute steering. This is supported by our CFG-vs-LoRA ablation (Table 13) and characteristic-specific augmentation gains (Table 4).
>
> 5. **Justification for choosing LoRA rank r=32 and scaling α=1.0** :
> We acknowledge the recent work showing task-specific sensitivity of LoRA hyperparameters. In this submission, we fixed the LoRA rank and scaling to a standard, stable setting (r=32, α=1.0) to limit degrees of freedom and keep comparisons consistent across attributes; subtlety is the only case where we intentionally vary α to provide a controllable strength knob. While a systematic sweep of rank/scale is interesting, we leave a full hyperparameter study to future work.
>
> 6. **Synthetic-only downstream training feasibility**: We did not try a synthetic-only downstream training experiment , Tables 3/5 evaluate real data supplemented with synthetic nodules. We expect its performance to be primarily limited not by background realism, but by (i) potential distribution gaps between synthetic vs. real nodule, and (ii) missing co-occurring real world factors (e.g., other pathologies/clinical confounders) that are naturally present when training on real positives.
>
> 7. **Combining multiple generator sources for augmentation (Table 5: mix ACGAN/ReACGAN/CRFill/DiT-XL vs single-source)**: We did not run an experiment that *mixes* synthetic samples from multiple generators in a single training set (e.g., 10k real + ACGAN + ReACGAN + CRFILL + DiT-XL), nor the variant excluding DiT-XL. Table 5 evaluates each generator as an alternative augmentation source under a matched setting (10k real + 10k synthetic), to keep the comparison controlled and attributable to the generator choice.
>
> Why we didn’t combine them:
> - Mixing generators introduces an extra variable: the downstream performance becomes a function of mixture weights and relative quality/diversity of each method, making it difficult to attribute gains or draw clean conclusions about which generator is responsible. Table 5 is designed to isolate this factor by holding the synthetic count fixed per method.
> - Also, several baselines (GAN/CRFill) can produce method-specific artifacts; mixing could either help via diversity or hurt by injecting more shortcut cues.
>
> What we can say :
> - Based on our scaling results (Table 3), we already see non-monotonic trends with increasing synthetic volume, suggesting there may be an optimal point rather than “more is always better.”
> - We can add this as a future experiment direction: exploring mixtures of generators and quality-vs-diversity trade-offs under a fixed total synthetic budget.
> 8. **Code/checkpoints (post-acceptance)**  - https://github.com/shreshthasingh00/Nodule-Crafter-Diffusion-driven-Nodule-synthesis-on-CXR/

---

> ### Comment · Reviewer_uH83 · 2026-01-30
>
> I thank the authors for addressing the feedback for all reviewers.
>
> A few side notes and some important points:
>
> > LoRA–attribute mapping clarification
>
> I see. Thanks for the clarification. For me, this was a bit hard to read from the figures and manuscript (unless I missed a detailed section on this, apologies for that). I suggest the authors address this in the manuscript for clarity (if not already taken care of).
>
> > Why detection gains plateau with more synthetic data:
>
> Thanks for addressing this. Any chance you could add this into the discussion section for the manuscript? (apologies if I missed it)
>
> > “LoRA vs no-LoRA/full finetuning” motivation
>
> This is great. Thanks for adding a dedicated section for this.
>
> > Motivation for LoRA beyond parameter efficiency (why LoRA vs classic finetuning; CFG vs LoRA control)
>
> Thanks again for the clarification.
>
> > Justification for choosing LoRA rank r=32 and scaling α=1.0
>
> _"We acknowledge the recent work showing task-specific sensitivity of LoRA hyperparameters. "_
>
> Could you please reference the work you state in the comment above?
>
> Although I agree that it's a great choice of parameters (again, only as per literature), it's a demanding systematic sweep necessary to begin with a control experimental setting. I don't wanna sound obsessed about this, but if it turns out at some point that a rank higher or lower than the 32 (which currently is not well motivated, and a hard-coded hyperparameter choice) works better for the proposed method, this would be a misleading portrayal (but I agree that it's definitely a lot of experiments for the rebuttal time at hand).
>
> > Synthetic-only downstream training feasibility
>
> Nice, thanks for sharing your hypothesis. It's quite interesting!
>
> > Combining multiple generator sources for augmentation (Table 5: mix ACGAN/ReACGAN/CRFill/DiT-XL vs single-source)
>
> Gotcha. Thanks for the response.
>
> > Why we didn’t combine them:
> > - Mixing generators introduces an extra variable: the downstream performance becomes a function of mixture weights and relative quality/diversity of each method, making it difficult to attribute gains or draw clean conclusions about which generator is responsible. Table 5 is designed to isolate this factor by holding the synthetic count fixed per method.
> > - Also, several baselines (GAN/CRFill) can produce method-specific artifacts; mixing could either help via diversity or hurt by injecting more shortcut cues.
>
> Ah true. This didn't cross my mind upfront. Thanks for elaborating and explaining your rationale.
>
> > Code/checkpoints (post-acceptance) - https://github.com/shreshthasingh00/Nodule-Crafter-Diffusion-driven-Nodule-synthesis-on-CXR/
>
> Hmm, this is the one aspect where I personally am not willing to take a conditioned answer on. There's only limited ways of ensuring reproducibility of new DL methods developed, and this is a key feature of MIDL to ensure reproducible/usable methods in deep learning. I'm sorry, but I am not happy about this (unless the authors have a strong rationale / hunch to choose against open-sourcing). And I didn't hear any response on releasing the in-house dataset curated for the study (on which the proposed methods were trained on).
>
> Overall, the authors have done a decent job addressing most of my questions, but I would appreciate if they could complement their manuscripts with the points raised (as most of my points were in direct context of the manuscript and/or the methods developed), or provide a fair rationale as to why they decided otherwise.

---

### Author Rebuttal · Authors · 2026-01-25

**Rebuttal:**

Revisions implemented (rebuttal version):
1. Corrected formatting inconsistencies and typographical errors throughout the manuscript.
2. Addressed all points raised during the rebuttal and incorporated the corresponding clarifications and updates.
3. Included the link to the code repository.
4. Highlighted all newly added or substantially revised sections in blue to facilitate review.

**Supporting Material:**

/attachment/53239a633b167ea92cc06be2f6550064f5f92537.pdf

---

### Meta-Review · Area_Chair_pKgq · 2026-02-07

**Recommendation:** Accept (Poster)
**Confidence:** 3

**Metareview:**

This paper presents a clinically relevant and well‑motivated contribution. The proposed diffusion‑based framework for fine‑grained nodule synthesis on chest radiographs addresses a real challenge in lung nodule detection: the lack of diverse, high‑quality annotated data.

The method is technically novel in the context of CXR nodule generation, particularly through the use of characteristic‑specific LoRA adapters for radiologically meaningful attributes and the orthogonality‑constrained merging strategy to combine them without interference.

Across reviewers, the overall reception was positive. The main concerns centered on the need for clearer justification behind some methodological choices (e.g., LoRA rank and scaling, the orthogonality loss), missing ablations, reproducibility issues, and figure or formatting inconsistencies.
The rebuttal addressed these points thoroughly. The authors clarified the attribute‑specific LoRA design, the reasoning behind hyperparameter choices, and the behavior of the orthogonality constraint. They also provided additional analysis, updated appendices, and included new tables that better highlight the effect of the orthogonality strategy.

One reviewer requested further experiments to move their score higher, but the authors’ explanations were thorough and reasonable, given the scope of the rebuttal period. In my opinion, the clarifications provided are sufficient, and the paper’s strengths outweigh the remaining open points. I think that the paper now contains sufficient clarification and information to support acceptance.

---

### Decision · Program_Chairs · 2026-02-13

Accept (Poster)